# Microfluidics-Based Biosensing Platforms: Emerging Frontiers in Point-of-Care Testing SARS-CoV-2 and Seroprevalence

**DOI:** 10.3390/bios12030179

**Published:** 2022-03-17

**Authors:** Elda A. Flores-Contreras, Reyna Berenice González-González, Iram P. Rodríguez-Sánchez, Juan F. Yee-de León, Hafiz M. N. Iqbal, Everardo González-González

**Affiliations:** 1Tecnologico de Monterrey, School of Engineering and Sciences, Monterrey 64849, Nuevo León, Mexico; eldafc@tec.mx (E.A.F.-C.); reyna.g@tec.mx (R.B.G.-G.); 2Laboratorio de Fisiología Molecular y Estructural, Facultad de Ciencias Biológicas, Universidad Autónoma de Nuevo León, San Nicolás de los Garza 66455, Nuevo León, Mexico; iram.rodriguezsa@uanl.edu.mx; 3Delee Corp., Mountain View, CA 94041, USA; juan.felipe@delee.bio

**Keywords:** SARS-CoV-2, COVID-19, microfluidic, chip, biosensors, diagnostics

## Abstract

Severe acute respiratory syndrome coronavirus 2 (SARS-CoV-2) caused the ongoing COVID-19 (coronavirus disease-2019) outbreak and has unprecedentedly impacted the public health and economic sector. The pandemic has forced researchers to focus on the accurate and early detection of SARS-CoV-2, developing novel diagnostic tests. Among these, microfluidic-based tests stand out for their multiple benefits, such as their portability, low cost, and minimal reagents used. This review discusses the different microfluidic platforms applied in detecting SARS-CoV-2 and seroprevalence, classified into three sections according to the molecules to be detected, i.e., (1) nucleic acid, (2) antigens, and (3) anti-SARS-CoV-2 antibodies. Moreover, commercially available alternatives based on microfluidic platforms are described. Timely and accurate results allow healthcare professionals to perform efficient treatments and make appropriate decisions for infection control; therefore, novel developments that integrate microfluidic technology may provide solutions in the form of massive diagnostics to control the spread of infectious diseases.

## 1. Introduction

The World Health Organization (WHO) has reported that, up to January 2022, the number of confirmed COVID-19 infections exceeded 340 million cases worldwide. Moreover, consequences in terms of public health have been severely exacerbated due to the viral evolution over time, which is causing the emergence of new variants. These mutations can give rise to substitutions in the amino acids of the viral proteins, affecting properties such as the propagation of the virus and the severity of symptoms post-infection, causing changes in public health measures and a loss in the effectiveness of drugs, vaccines, and diagnostic methods [1].

Therefore, massive diagnostic tests for detecting SARS-CoV-2 have been suggested by the WHO to decrease the spread of the virus and its consequences on populations’ health and the global economy. In this manner, accessible, affordable, and accurate diagnostic devices are urgently required for SARS-CoV-2 and its emerging variants, both for controlling the spread of COVID-19 and for the resumption of economic activities [2]. In this regard, quantitative reverse-transcription–polymerase chain reaction (RT-qPCR) has been reported to be the most reliable method for the screening and diagnosis of COVID-19 [3,4], and is the most predominantly used method through the analysis of respiratory samples [3,4]. Although this method is highly sensitive, it requires expensive instruments and reagents, extensive sample processing, and specialized personnel, limiting its large-scale application outside equipped clinical laboratories [2,5].

To overcome these challenges, microfluidics for point-of-care (POC) testing devices have been recently applied for COVID-19 [2,6]. In this work, we review and discuss the different types of microfluidic-based tests relevant to COVID-19. We divide the tests into three categories based on their detection targets: nucleic acid, antigens, and anti-SARS-CoV-2 antibodies. This review examines the emerging role of microfluidics in a pandemic context that demands a reduction in costs and the substitution of sophisticated equipment, massifying and democratizing diagnosis for the entire population.

## 2. Microfluidics Applied to COVID-19

Microfluidics is an emerging technology that studies the behavior, control, and manipulation of fluids, either gases or liquids, constrained into micrometer- or nanometer-sized structures or channels [7]. The behavior of the fluids in such microchannels is different in comparison to the macroscale. Consequently, microfluidic systems exhibit apparent advantages such as: (i) less sample is required, (ii) lower consumption of reagents, (iii) the ability for simultaneous sample processing, (iv) accelerated reaction rates, (v) enhanced precision and sensitivity, and (vi) reductions in the quantities of waste products [8,9].

Due to the ability of microfluidics to accurately handle small quantities of samples and its use of well-controlled environments, it has been effectively applied in different applications such as nanofabrication [10], energy generation [11], and optofluidic reactors for water treatment [12]. Moreover, microfluidic systems have been successfully applied as diagnostic devices, motivated by the inaccessibility, impracticality, unaffordability, and high costs of fully equipped laboratories for a large part of the population [13].

Microfluidics forms the basis of technologies such as lab-on-a-chip, which allows any sample analysis procedure—separation, concentration, dosing, mixing, incubation, reaction, and detection—to be integrated on a microfluidic chip [9]. Moreover, the operation of microfluidic platforms can be completely automated, thus reducing human errors and improving repeatability [14]. In this sense, microfluidic research on diagnosis has been rapidly increasing in recent times (Figure 1).

Microfluidics diagnostic systems are integrated by detection components and fluid regulatory elements that can detect different molecules such as nucleic acid, antigens, or antibodies, producing an optical, electrical, or electrochemical signal with a diagnostic result [15] (Figure 2). In addition, their portability, convenience, and connectivity allow for the rapid examination of diagnostic results near the patient site (point-of-care devices) [16]. Therefore, research efforts in microfluidics have been focused on the diagnosis of different infectious diseases, such as dengue [17,18], Malaria [19], Zika [18], and more recently, SARS-CoV-2 [2].

## 3. Nucleic Acid Detection

The first sequence of the SARS-CoV-2 virus was obtained at the end of 2019. This achievement enabled the possibility of accessing its genetic code in databases, which is the most critical information for developing diagnostic techniques based on nucleic acid detection. SARS-CoV-2 is a single-stranded positive RNA virus with several genes such as the ORF1ab, RdRP, E, N, and S genes. The first diagnostic tests approved and used in this pandemic were designed to detect the N gene. Currently, researchers have developed different tests with multiple-gene detection to increase the certainty and ability to identify viral variants [20,21]. RT-qPCR is the gold standard for COVID diagnosis and the test recommended by the WHO due to its relatively elevated sensitivity and specificity. In addition, RT-qPCR tests have been extensively studied, showing significant advances in understanding the technology and infrastructure necessary to implement protocols worldwide [22]. However, the pandemic requires better techniques to manage the outbreak efficiently; thus, research efforts are imperative to develop new diagnostic tools that meet the current demands. We present several proposals regarding nucleic acid detection by means of microfluidics, divided into three main sections depending on the technology used: PCR, isothermal amplification, and clustered regularly interspaced short palindromic repeats (CRISPR)/biosensors. Figure 3 presents a graphical overview of microfluidic techniques used to detect nucleic acid from SARS-CoV-2.

### 3.1. Based on PCR

Millions of RT-qPCR tests have been performed worldwide, which have significantly helped to diagnose COVID-19. However, the severity of the pandemic makes the traditional PCR method insufficient to control the problem. The most important limitations are shortages of reagents, limited staff, the analysis time, and the massive sample processing requirements [22]. Several proposals have recently been reported in scientific journals to solve those challenges. For example, a portable microfluidic-chip qPCR method that can detect the SARS-CoV-2 virus directly from saliva samples has been recently reported. This technique utilized a microfluidic chip where the sample is placed, and all the reaction stages (heat, mix, and detection) are performed; this device can perform multiplex assays detecting up to three targets, showing a virus detection of 1000 copies/mL in saliva samples [23].

An essential approach in microfluidics is high-throughput sample processing. Fassy et al. and Xie et al. have reported the use of a qPCR method with the potential to perform massive assays to detect the SARS-CoV-2 virus [24,25]. The reported technology is based on a nanofluidic qPCR method using the 192.24 IFC chip (Fluidigm, CA, USA) (Figure 3), which can process 4608 samples in a single run. This technology also performs multiplex assays with 24 different probe sets, which is very convenient for identifying SARS-CoV-2 variants or confirming the diagnosis with the amplification of other targets. Moreover, the reaction volume was reported to be as low as 10 nL, which is an important characteristic in terms of reducing the amount of reagents used and the consequent decrease in the price of tests [24,25].

Using this technology, Fassy et al. proposed an optimized protocol for SARS-CoV-2 RNA detection without the need for viral RNA extraction, which reduces the processing time [24]. Through microfluidics, researchers have pursued volume reduction with the aim of producing low-cost tests. For example, Cojocaru et al. reported a PCR test based on a disposable microchip with lyophilized primers and probes capable of providing results in less than 30 min using a volume of 1.2 μL per reaction. They determined a limit of detection (LOD) of one copy per reaction with clinical samples from Canadian patients [26]. Moreover, another microfluidic form of PCR is the lab-on-a-disc, applied to COVID-19, including the multi-detection of Influenza A and B. The LOD of this device was 20 viral copies, with a test time of 1.5 h [27]. Another recent and novel development is a microfluidic nanoplasmonic qPCR method based on a microfluidic chip that integrates glass nanopillar arrays with Au nanoislands and microfluidic channels, in which the PCR reaction amplifies the E gene from SARS-CoV-2. The main characteristic of the reported microfluidic system is its rapid capacity to detect the presence of SARS-CoV-2, requiring around five minutes with an LOD of 259 copies/μL [28]. Overall, these microfluidic PCR platforms have demonstrated the potential for optimal and rapid virus detection.

In addition, one of the objectives of microfluidics is to integrate molecular techniques in microfluidic chips, developing complete laboratories on a chip that provide relevant information for a more precise diagnosis. Li et al. [29] reported the synergistic potential of combined technologies and proposed a workflow using a microfluidic platform to detect SARS-CoV-2 and then performing whole-genome sequencing. These devices could be responsible for controlling the pandemic in the future due to the possibility of increasing the sensitivity of the tests and reducing the costs and time of the tests [29].

Digital PCR (dPCR) is an emergent and highly precise PCR test, which has received increasing research attention since it provides absolute quantification of nucleic acid molecules from very small samples [30]. The most crucial difference between traditional qPCR and dPCR is the reaction preparation and the equipment required; dPCR requires, as the first step, the preparation of the sample to be separated in the dispensed reaction, mainly by means of micropores and droplets. In this way, the overall response can be divided into multiple compartmentalized reactions, generating small reactors to amplify nucleic acid independently with a Poisson distribution, leading to analysis based on fluorescent signals in each compartment. dPCR has shown potential for its use in the clinical field. Thus, several research groups have employed it to diagnose COVID-19. Bu et al. developed a microfluidic system that generates programmable on-demand droplets, with the viral detection of 4.68 copies/μL, amplifying the ORF1ab and N genes [31]. A similar SARS-CoV-2 dPCR test was reported by Yin et al., which showed equivalent results in viral detection (five copies/reaction), amplifying the same two genes with the ability to detect the virus in less than 5 min [32] (Figure 3). Recently, Sun et al. reported a dPCR test composed of a microfluidic chamber manufactured using a wet-etching process and silicon-glass bonding. Some interesting features of this microfluidic chip are the low-cost fabrication method, the generation of microdroplets of uniform volume, and higher stability, with the authors reporting an LOD of 10 copies/μL, using the ORF1ab gene as the target [33].

### 3.2. Based on Isothermal Amplification

The use of a thermocycler is essential to perform PCR tests. However, infrastructure deficits in underdeveloped places is one of the most critical limitations. Therefore, there is an urgent need to develop strategies that do not require expensive equipment, thus achieving a diagnosis that is accessible to the entire population, regardless of their economic context. For example, there is an approach based on the isothermal amplification of nucleic acid, which avoids the use of a thermocycler, replacing it with a simple water bath or incubator. Several isothermal techniques have been developed and deployed, such as loop-mediated isothermal amplification (LAMP), rolling circle amplification (RCA), and recombinase polymerase amplification (RPA) [34]. The isothermal COVID-19 diagnosis method has attracted worldwide attention due to its potential to massify the diagnosis in places where there are no laboratories or expensive facilities.

LAMP is a technology that employs four to six primers to recognize different regions from a target sequence; it is distinguished by requiring a temperature reaction around 65 °C and having shorter reaction times compared to other techniques such as PCR. To date, the FDA has approved ten COVID-19 tests based on LAMP, and this technology has been widely applied for the development of COVID-19 microfluidic devices. One of the first developments was an automated microfluidic disc that is able to perform the entire process—sample treatment, LAMP reaction, and fluorescence signal detection—after injecting samples into the disc. The estimated LOD is two copies per reaction, enabling the processing of up to 21 samples per microfluidic disc in 70 min [35]. Another successful example of this technology is a device integrated within an aluminum block, detecting fluorescence emission using a camera controlled by a single-board computer. The authors demonstrated the ability to detect 100 copies of viral RNA within 10 to 20 min, estimating the device’s cost at around 150 USD [36]. Ganguli et al. created a platform that includes a microfluidic chip and a smartphone-based reader that demonstrated an LOD of 50 copies/μL within 30 min [37]. Similarly, de Oliveira et al. employed a cell phone as a reader; they printed a polyester-toner microfluidic device controlled by a fidget spinner using centrifugal force. The microfluidic chip contains chambers with a 5 μL of volume capacity, and to perform the LAMP reaction, the sample is incubated using a thermoblock. They reported a time assay of 10 min, with an LOD of 10^−3^ copies of viral RNA and an estimated cost of 5 USD per test [38].

Recently, colorimetric LAMP tests based on changes in pH values have been developed, which have also been implemented in microfluidics. Davidson et al. prepared microfluidic paper-based analytical devices (μPADs) to detect SARS-CoV-2 via colorimetry. They demonstrated direct viral detection from saliva samples (without sample processing), obtaining results that were visible to the naked eye. The microfluidic device could detect up to 200 copies/μL from saliva samples within 60 min, with an estimated cost of 10 USD per test [39]. Furthermore, Deng et al. developed a portable device that employs a colorimetric LAMP with an LOD of 300 copies/reaction within 35 min [40]. On the other hand, Kim et al. reported a microfluidic device using RCA, which is another isothermal technique that employs a short primer and is amplified to form long single-stranded nucleic acid using a circular template and polymerases. This test is based on a device integrated with a nylon mesh with multiple microfluidic pores and an immobilized primer. Its operation is based on the fact that RCA amplification occurs when the viral target is present, causing DNA gelation and, consequently, a blockage of the micropores, preventing the flow of microfluidics. This device was able to detect SARS-SoV-2 within 5 min at a concentration of 30 aM [41] (Figure 3).

The RPA isothermal assay has also been applied in microfluidic testing for COVID-19; this technology has interesting features such as high sensitivity, fast reaction times, lower temperatures (37 °C to 42 °C), and simple protocols and primer design. RPA is based on recombinase enzymes to facilitate primer binding to templates and the synthesis of strands. As in other methods, it is also possible to quantify the amplification process via fluorescence or colorimetry using lateral flow assays. For example, Liu et al. developed a microfluidic chip and combined RPA and lateral flow assays to diagnose COVID-19. They fabricated a PMMA (polymethyl methacrylate) microfluidic chip with two reservoirs, in which the reactants are mixed, and the reaction is conducted. It also includes a strip with the conjugated antibodies to perform the colorimetric detection. The chip is incubated at 42 °C for 15 min using a thermoblock; after the amplification time, the chip is inverted to start the reveal process with the lateral flow assay. They reported an LOD of 1 copy/μL within 30 min [42] (Figure 3). Interestingly, a microfluidic disc combining LAMP and RPA technologies was recently developed, demonstrating the ability to detect multiple targets of SARS-CoV-2 and measles virus via fluorescence, with an LOD as low as 10 copies within one hour of the reaction [43].

### 3.3. Based on CRISPR

CRISPR is an emerging and powerful technique that has also been applied to detect nucleic acid, exhibiting high versatility, sensitivity, and specificity. The use of CRISPR in diagnostics is based on the reaction of “collateral cleavage” induced by a nuclease, such as Cas12 and Cas13. These CRISPR-Cas nucleases can be directed to a guide RNA (gRNA)-specific target sequence. Furthermore, this reaction can be analyzed through fluorescence signals, a practical approach to detecting pathogens.

Several research groups have integrated CRISPR into microfluidic technology to detect SARS-CoV-2, developing novel devices to improve diagnosis. One of the first CRISPR microfluidic chips developments applied to the diagnosis of COVID-19 used an electrokinetic microfluidic technique termed “isotachophoresis” (ITP) (Figure 3). Using this technology, Ramachandran et al. developed a microfluidic chip to extract nucleic acid from raw biological samples. Furthermore, the chip can mix the reagents and accelerate enzymatic reactions with the viral target. The authors reported multiple benefits obtained using this technique, such as lower consumption of reagents in comparison to traditional methods (lower than 0.2 μL), a decrease in the sample processing time (30–40 min from sample to results), and an LOD as low as 10 copies/μL [44].

Li et al. proposed a microfluidic system combining RPA amplification, CRISPR cleavage, and lateral flow detection. Their developed microfluidic chip was integrated with a reaction chamber that stored lyophilized reagents and a portable hand warmer to incubate the reactions to amplify SARS-CoV-2 nucleic acid, avoiding the use of electricity; this portable microfluidic system displayed an LOD as low as 100 copies [45] (Figure 3). Similarly, a CRISPR assay that is able to detect SARS-CoV-2 was implemented in a microfluidic chip with reaction chambers, a heating module to incubate the reactions at 37 °C, and a compact fluorescence imaging system for monitoring the fluorescence signal; an LOD of 31 copies/μL within 20 min of reaction was reported [46].

### 3.4. Other Microfluidic Developments

The detection of SARS-CoV-2 nucleic acid has also been achieved using other types of microfluidic biosensors. For example, Zhao et al. developed an automated microfluidic platform based on nanotechnology to detect the S gene from the SARS-CoV-2 virus. This technology, known as the “electrochemical system integrating reconfigurable enzyme-DNA nanostructures” (eSIREN), integrates multiple responsive molecular nanostructures to form a catalytic molecular circuit with the aim of sensing the viral presence. It presented an LOD of 7 copies/μL after approximately 20 min of reaction at room temperature [2].

Another example of technologies not frequently explored is the one developed by Hwang et al., which consists of interdigitated platinum/titanium electrodes to detect SARS-CoV-2 nucleic acid. In this technique, the detection is based on sensing the hybridization from the viral analyte with probe DNA, and using physicochemical analytical techniques such as Fourier-transform infrared (FTIR) spectrometry, contact-angle analysis, and capacitance-frequency measurements. This approach allowed the detection of the RdRp gene with a sensitivity of 0.843 nF/nM [47]. Finally, Iwanaga reported the development of a microfluidic chip using a biosensor based on an all-dielectric metasurface fabricated with silicon-on-insulator nanorod arrays to enhance the fluorescence signal and demonstrated an LOD of 250 amol/mL within 30 min of the reaction [48]. Table 1 summarizes the diverse microfluidic assays used for the detection of SARS-CoV-2 nucleic acid.

## 4. Antigen Detection

Currently, there are a few reports on microfluidic immunoassays for the detection of SARS-CoV-2 viral proteins, despite their multiple benefits such as their low cost and the rapid processing of samples compared to PCR tests that use nucleic acid detection. In addition, antigen immunoassays can perform more convenient detection since they can detect the presence of the virus when the person can transmit it, unlike serological immunoassays that identify IgG and IgM antibodies against SARS-CoV-2, which require at least a few weeks for their detection. Therefore, immunoassays of antigens allow effective measurements of viral proteins, obtaining results in less than an hour with detection ranges from fg to µg (Table 2).

The most relevant antigens for detecting SARS-CoV-2 in clinical samples are the nucleocapsid protein (N) and the spike protein (S). The S protein is a glycoprotein on the virus’s surface, composed of S1 and S2. In addition, S1 contains a region known as the receptor-binding domain (RBD), which facilitates the entry of SARS-CoV-2 into the host cells [49,50]. The N protein is an immunogenic protein that packages the genome RNA of the virus, forming helical nucleocapsids [50,51]; it is used as an early indicator since it allows the identification of SARS-CoV-2 up to one day before symptoms appear [52]. Furthermore, the N protein is considered a better target than the S protein because the latter is less abundant and, under selective pressure, is more prone to mutations [53]. The detection of SARS-CoV-2 proteins by means of microfluidic immunoassays consists mainly of flow-through assays [54]. The sample obtained from serum or nasopharyngeal or oropharyngeal swabs [55] is subjected to this flow to identify viral SARS-CoV-2 proteins via direct or sandwich immunoassays. Direct immunoassays consist of immobilized anti-SARS-CoV-2 antibodies or nanomaterials (polymers and fibronectin) on a sensor’s surface, which allows one to identify the presence of proteins such as S, RBD, and N [56,57,58,59]. In contrast, sandwich-type immunoassays consist of the formation of complexes between anti-SARS-CoV-2 antibodies anchored to microspheres, nanobeads, or microbeads that interact with SARS-CoV-2 viral proteins such as S, RBD, and N. These complexes are subsequently captured by aptamers or a second anti-SARS-CoV-2 antibody, either immobilized (on a gold electrode or silica) or free (attached to fluorescent reporters or colored nanobeads) [53,57,60,61]. The identification of SARS-CoV-2 virus in samples of interest by means of direct or sandwich microfluidic immunoassays is performed after the interaction of the anti-SARS-CoV-2 antibodies, nanomaterials, or aptamers with the viral proteins, which produes a fluorescence signal, or changes in electric current, absorbance, or color [62,63,64].

The primary type of microfluidic immunoassays is based on the interaction of viral proteins with anti-SARS-CoV-2 antibodies or nanomaterials immobilized on a sensor (which detects changes in the electric current) or an electrode, typically covered with graphene, aluminum, fluorine-doped tin oxide electrode (FTO), screen-printed carbon electrode (SPE), or gold nanoparticles (AuNPs). These materials allow the identification of electrical conductivity changes when the antibodies interact with the antigens [57,59,65]. As a representative work of this type of immunoassay, Seo et al. developed a direct microfluidic immunoassay that uses a field-effect transistor (FET) coated with graphene sheets conjugated with anti-SARS-CoV-2-Spike antibodies that cause a voltage change when interacting with the viral particles. This chip did not require the processing or labeling of samples obtained from a nasopharyngeal swab and presented an LOD of 2.42 × 10^2^ copies/mL in less than 5 min [66]. Another important type of microfluidic immunoassay is generating a redox reaction caused by anti-SARS-CoV-2 antibodies conjugated with enzymes (that interact with their substrate) or redox probes (electron transfer) in the presence of the SARS-CoV-2 viral proteins. Li and Lillehoj reported an immunoassay using this type of signal, which consumes minimum quantities of reagents and enhances the detection sensitivity. Their immunosensor comprises a novel sandwich immunoassay. It uses dually-labeled magnetic nanobeads tagged with HRP (horseradish peroxidase). It is covered with anti-SARS-CoV-2 antibodies that interact with the viral proteins, forming a complex captured by a second anti-SARS-CoV-2 antibody immobilized on a gold electrode. In the presence of a TMB substrate, this complex produces an electrochemical signal that can be detected by smartphones or electronic devices, with an LOD of 50 pg/mL in whole serum samples, providing results in less than an hour [60].

Interestingly, Yousefi et al. designed a direct immunoassay that integrates an antibody conjugated with a negatively charged DNA linker, labeled with a redox probe (ferrocene), which binds to the surface of the positively charged sensor when it interacts with an S protein, thus generating a redox reaction that the sensor detects. This immunosensor has an LOD of 4 × 10^3^ viral particles/mL in unprocessed saliva samples, generating results in approximately five minutes [58]. On the other hand, Raziq et al. proposed an electrochemical sensor using molecularly imprinted polymers (MIPs) acting as the antibodies; they were interfaced with a thin film electrode and connected to a portable potentiostat, providing high sensitivity and the ability to discriminate between molecules to exclusively detect N proteins from nasopharyngeal samples with an LOD of 15 fM [63].

Microfluidic immunoassays that use fluorescence signals to detect SARS-CoV-2 consist of antibodies or aptamers labeled with fluorescent reporters (probes, fluorophores, or submicron particles) that emit a signal that is directly proportional to the viral particles dispersed in the analyzed sample [56,67]. Stambaugh et al. presented a successful example of this technique, creating a photonic chip-based sandwich immunoassay. Basically, the anti-SARS-CoV-2 antibody is linked to a fluorescent DNA probe that has a photo-cleavable spacer, which interacts with the SARS-CoV-2 viral proteins, forming a complex. This complex is captured by a second antibody, which emits a fluorescence signal when is exposed to UV radiation. This compact device allows the simultaneous detection of Influenza A and SARS-CoV-2 with an LOD of 30 ng/mL from nasopharyngeal samples [68]. Alternatively, Ge et al. developed a test for the effective detection of the SARS-CoV-2 virus at the femtoliter scale. The device confined magnetic beads covered with anti-SARS-CoV-2-N antibodies and biotin-labeled aptamers interacting with the N protein, forming a sandwich immunoassay. The fluorescence signal was observed in the presence of the SARS-CoV-2 virus and streptavidin-B-galactosidase. This device had an LOD of 33.28 pg/mL, which is 300 times lower than that of the traditional ELISA sandwich method [61].

Optical techniques have also been used to detect SARS-CoV-2 proteins, in which antibodies anchored to AuNPs, enzymes, or colored nanobeads are used; their interaction with viral proteins leads to changes in the absorbance or the observation of a color signal in the visible spectrum. An excellent example of this technique was reported by Xu et al., who prepared a handheld microfluidic hydrodynamic filtration device based on a sandwich immunoassay, in which the N protein of SARS-CoV-2 interacts with anti-SARS-CoV-2-N antibodies, binding to white microbeads and to a second antibody anchored to a red nanobead, forming a complex. Then, the complex enters the observation zone (OZ) via microfluidic filtration; a red color is observed when the antigen is present, whereas a white color is observed in the absence of the antigen since free red nanobeads pass through the pillar gaps, leaving the white beads in the OZ. This device had an LOD of <100 copies/mL in nasal samples, with outstanding sensitivity and specificity of 95.4% and 100%, respectively. In addition, they reported an estimated cost of 0.98 USD per test and the possibility of reusing the device more than 50 times [53].

Regarding the use of conjugated enzymes, Sun et al. proposed a paper-based microfluidic sandwich immunoassay, which consists of an antibody conjugated to HRP and a second anti-SARS-CoV-2-N antibody immobilized on the chitosan-glutaraldehyde surface of the immunosensor. In the presence of the N protein, this biosensor generates a black spot observed with the naked eye with an LOD of 8 µg/mL [64]. On the other hand, Murugan et al. reported a biosensor based on absorbance changes. The authors created a plasmonic fiber-optic absorbance biosensor (P-FAB) on a U-bent optical fiber probe. In their work, two different types of design were proposed: a direct immunoassay and a sandwich immunoassay. In the direct immunoassay, antibodies were immobilized to AuNPs placed on the biosensor’s surface. In contrast, the sandwich immunoassay consisted of antibodies anchored to the biosensor’s surface and secondary antibodies conjugated with AuNPs. In both designs, the interaction of the N proteins with the antibodies resulted in absorbance changes. This portable device had an LOD of 10^−8^ M for saliva samples (using a volume of 25 µL) with minimal pre-processing procedures, obtaining results within 15 min [62]. These types of microfluidic immunoassays allow the rapid detection of SARS-CoV-2 proteins (<1 h) and involve minimum or null sample processing, compared to ELISA. However, this traditional technique involves time-consuming processes ranging from several hours to days. Another emergent method to detect viral antigens is through Raman spectroscopy. Recently, Huang et al. reported a rapid assay combing Raman spectroscopy and a deep learning model to detect the S protein from COVID-19 patients within 20 min [69].

**Table 2 biosensors-12-00179-t002:** Microfluidic immunoassays for the detection of SARS-CoV-2 proteins.

Type Immunoassay	Specimen	LOD	Target Protein	Detection Method	Processing Time (Minutes)	Reference
Sandwich Immunoassay	Serum	33.28 pg/mL	N	Fluorescence	<120	[61]
Direct Immunoassay	Saliva	NR	VP	Fluorescence	<30	[56]
Sandwich Immunoassay	Serum, Saliva, Nasopharyngeal and urine	8 µg/mL	N	Colorimetric	>30	[64]
Direct and sandwich Immunoassay	Saliva	NR	N	Absorbance	15	[62]
Direct Immunoassay	Blood	1 fg/mL	S	Voltage	0.05	[57]
Direct Immunoassay	Saliva	4000 viral particles/mL	S	Electrochemical	5	[58]
Direct Immunoassay	Food	2.29 × 10^−6^ ng/mL	S	Voltage	0.33	[59]
Direct immunoassay	Saliva	90 fM	S	Electrochemical	0.5	[65]
Sandwich immunoassay	Serum	230 pg/mL	N	Electrochemical	<60	[60]
Sandwich immunoassay	Nasopharyngeal and serum	NR	VP	Fluorescence	15	[67]
Sandwich immunoassay	Nasopharyngeal	<100 copies/mL	N	Colorimetric	>30	[53]
Sandwich immunoassay	Nasopharyngeal	30 ng/mL	N	Fluorescence	<120	[68]
Direct immunoassay	Nasopharyngeal	2.42 × 10^2^ copies/mL	S	Voltage	>1	[66]
Direct immunoassay	Serum	1 pg/mL	S	Voltage	15	[70]
Direct immunoassay	Nasopharyngeal	15 fM	N	Electrochemical	>30	[63]

Abbreviations: LOD (limit of detection); N (nucleocapsid protein); S (spike protein); VP (unspecified SARS-CoV-2 viral proteins); NR (not reported).

## 5. Anti-SARS-CoV-2 Antibody Detection

Serological tests for SARS-CoV-2 are highly relevant since they not only provide information on the existence of a viral infection but also on the severity of the disease (which is correlated with age and antibody expression), as well as information on the success of vaccination in people not infected with SARS-CoV-2 [71,72]. Antibody expression occurs as a reaction to the SARS-CoV-2 virus, secreted by B lymphocytes. There are various antibodies involved; however, IgG and IgM isotypes are the most relevant for developing serological tests since these isotypes are found more frequently in the blood and are expressed during different periods [73,74,75]. IgM antibodies appear between the fourth and tenth day post-infection, whereas IgG antibodies are expressed during the second week post-infection [74,76,77,78]. Microfluidic immunoassays for detecting antibodies against SARS-CoV-2 consist mainly of antigens (S, RBD, or N proteins) that capture IgM or IgG antibodies depending on the post-inoculation time of SARS-CoV-2 via direct, indirect, and sandwich immunoassays (Table 3) [79,80,81].

Direct immunoassays consist of an immobilized antigen on a sensor or electrode that captures IgG or IgM antibodies. In contrast, indirect immunoassays consist of an antigen immobilized on the surface of the microfluidic chip, sensor, or electrode (glass/polydimethylsiloxane, polystyrene, silicon, or paper) that binds to IgG and IgM antibodies. However, it is necessary to use a second antibody, either label-free or labeled by an enzyme or fluorophore, in the microfluidic environment emitting a signal (fluorescence, voltage change, electrochemical, or colorimetric) in the presence of antibodies against the SARS-CoV-2 virus [81]. On the other hand, sandwich-type immunoassays consist of an antigen fixed on the surface of the microfluidic device. They require a second antigen that is labeled (not immobilized); if the antibodies are against SARS-CoV-2, they will interact with both antigens producing a fluorescent signal [82].

The signals produced by microfluidic immunoassays to detect anti-SARS-CoV-2 antibodies (IgG or IgM) include fluorescence, colorimetry, and electrochemical signals. Thus, microfluidic immunoassays can be classified as label-based or label-free assays [83,84,85]. Label-based microfluidic biosensors for detecting antibodies are usually indirect or sandwich immunoassays. They typically consist of secondary antibodies (anti IgG or IgM) linked to enzymes (in the presence of their substrates) and fluorophores producing color changes and fluorescence signals, respectively, when they interact with antibodies against SARS-CoV-2. They can reach low LODs in the range of pg/mL to ng/mL, providing results in less than one hour [82,86,87].

Regarding microfluidic assays based on colorimetry changes, González-González et al. reported an automated ELISA on-chip of polystyrene with four straight channels (with a capacity of 50 µL/channel). This device is based on an indirect immunoassay that is able to detect anti-SARS-CoV-2 antibodies against the S protein from serum samples, either from vaccinated or COVID-19 patients. This microfluidic device consists of S proteins immobilized on the surface of a microfluidic chip that, using a secondary antibody labeled with HRP (in the presence of the substrate TMB), detects anti-SARS-CoV-2 (IgG) antibodies. A colorimetric signal is visible through the use of a smartphone or a microplate reader. In addition, processes can be programmed using Zen lab software, avoiding sample manipulation and human error [80]. Tripathi and Agrawal developed a semi-automated on-chip ELISA to detect anti-SARS-CoV-2 antibodies and were able to separate 10 µL of serum from 1 mL of whole blood in approximately 3 min. They used a microfluidic chip fabricated out of polydimethylsiloxane (PDMS) and glass covered with the S protein [81].

Tan et al. developed a microfluidic immunoassay that detects anti-SARS-CoV-2 (IgG) antibodies using portable microfluidic chemiluminescent ELISA technology. The device is made of polystyrene and has 12 channels, with an inner diameter of 0.8 mm, requiring only 8 µL of serum. This immunoassay consists of an anti-polyhistidine antibody that immobilizes the S and N proteins that interact with anti-SARS-CoV-2 antibodies, which in turn bind to the secondary antibody (anti-IgG) labeled with HRP. This immunosensor has an LOD of 0.06 ng/mL and 1 ng/mL for the N and S proteins, respectively. This microfluidic chip also allows the detection of antigens (S and N proteins) by immobilizing a captured antibody on the surface of the chip [88].

On the other hand, the devices that use fluorescence signals to identify anti-SARS-CoV-2 antibodies consist of antigens or secondary anti-IgG or anti-IgM antibodies labeled with fluorophores (phycoerythrin and Dylight 550). An example of their successful application for the detection of SARS-CoV-2 is a polydimethylsiloxane (PDMS) microfluidic chip developed by Lee et al., which comprises carboxylate polystyrene beads that immobilize the RBD protein on the microfluidic chip and conjugated secondary antibodies with Dylight 550 (not immobilized) that release a fluorescence signal when they bind to anti-SARS-CoV-2 antibodies. Furthermore, this microfluidic chip allows the detection of Zika, Dengue, and Chikungunya viruses within 30 min [86]. Another example of fluorescence immunoassays was proposed by Rodriguez-Moncayo et al., who used a multiplex format on a semi-automated platform made of PDMS/glass that allows the identification of the affinity of anti-SARS-CoV-2 antibodies (IgG or IgM) towards different viral proteins (S, N, RBD, and S1). This device can process up to 50 samples with an LOD of 1.6 ng/mL, and a sensitivity and specificity of 95% and 91%, respectively [87].

Similarly, Heggestad et al. reported a microfluidic immunoassay that determines the affinity of different SARS-CoV-2 viral proteins (S1, N, and RBD). They used labeled antigens with fluorophores on a platform that consisted of a completely autonomous immunoassay. All the reagents were added to the surface of the platform via inkjet printing. The reported sensitivity of this device was 100% for antibodies that recognize S1 and RBD proteins and 96.3% for antibodies that interact with the N protein. The specificity for the S1, RBD, and N proteins was 100% two weeks after the appearance of symptoms [82].

On the other hand, label-free microfluidic assays detect changes in the electrical current and refractive index. They are typically used in direct or indirect immunoassays to detect the presence of anti-SARS-CoV-2 antibodies (IgG or IgM). These types of immunoassays have an LOD for antibodies (IgG or IgM) against SARS-CoV-2 in the range of ng/mL in diluted serum samples. Qualitative microfluidic chips have also been reported, indicating only the presence of anti-SARS-CoV-2 antibodies. Sample processing and collection for label-free-type immunoassays require less than an hour [84,89,90]. An example of this type of biosensor was proposed by Li et al., consisting of a direct immunoassay on a working electrode of pieces of paper and carbon ink and zinc oxide nanowires (ZnO NWs), with an LOD of 10 ng/mL in serum samples. The RBD protein was immobilized, detecting a current change via electrochemical impedance spectroscopy (EIS) when it interacts with the anti-SARS-CoV-2 antibodies (IgG) [84]. Another novel immunoassay was reported by Djaileb et al., detecting IgG antibodies specific for the N and S proteins of infected or vaccinated patients in serum, plasma, and dried blood spot samples. The immobilized viral proteins on the surface of the plasmon resonance instrument produced a shift in the refractive index when they interacted with anti-SARS-CoV-2 antibodies. The LOD was 2 nM for indirect immunoassays and 3 nM for direct immunoassays [89].

Funari et al. developed an indirect immunoassay based on a localized surface plasmon in an opto-microfluidic chip. The S protein was immobilized on gold nanostructures, causing a change in the refractive index after interacting with anti-SARS-CoV-2 (IgG) antibodies and a secondary anti-IgG antibody. This immunoassay has an LOD of approximately 0.08 ng/mL (0.5 pM), and the results are obtained in 30 min [79]. Xu et al. developed an all-fiber Fresnel reflection microfluidic (FRMB) that detects IgG and IgM antibodies against the S protein through a secondary antibody (anti IgG or IgM) acting as a transducer and biorecognition element. The quantification is achieved according to the relationship of the intensity of Fresnel reflection light. This biosensor provided results in 7 min and presented an LOD of 0.82 ng/mL for IgM and 0.45 ng/mL for IgG antibodies [90].

Overall, these microfluidic chips have a lower LOD and higher specificity than traditional ELISA methods [91]. Furthermore, these devices are highly useful for evaluating the success of vaccination programs, estimating the affinity of anti-SARS-CoV-2 antibodies for new variants, and finding plasma donors for patients severely affected by SARS-CoV-2. However, there are still limitations, such as the production time required for the antibodies, on which the sensitivity depends, to avoid false negatives. Therefore, further research is still required to achieve accurate results and practical application.

**Table 3 biosensors-12-00179-t003:** Microfluidic immunoassays for the detection of anti-SARS-CoV-2 antibodies.

Type Immunoassay	Specimen	LOD	Target Antibodies	Detection Method	Processing Time (Minutes)	Reference
Indirect immunoassay	Serum	NR	Anti-S	Colorimetric	<150	[80]
Sandwich immunoassay	Blood	NR	Anti-S	Colorimetric	<5	[81]
Indirect immunoassay	Serum	0.06–1 ng/mL	Anti-N and S	Chemiluminescent	15	[88]
Indirect immunoassay	Serum, nasopharyngeal	NR	Anti-RBD	Fluorescence	30	[86]
Indirect immunoassay	Serum	1.6 ng/mL	Anti-N, S and RBD	Fluorescence	<90	[87]
Sandwich immunoassay	Blood	0.12 ng/mL	Anti-N, S and RBD	Fluorescence	60	[82]
Direct immunoassay	Serum	10 ng/mL	Anti-RBD	Electrochemical	30	[84]
Direct and indirect immunoassay	Blood	2–3 nM	Anti-N and S	Absorbance	30	[89]
Indirect immunoassay	Blood	0.08 ng/mL	Anti-S	Absorbance	30	[79]
Indirect immunoassay	Serum	0.82–0.45 ng/mL	Anti-S	Absorbance	7	[90]
Indirect immunoassay	Serum	NR	Anti-S and RBD	Absorbance	NR	[83]

Abbreviations: LOD (limit of detection); N (nucleocapsid protein); S (spike protein); RBD (receptor-binding domain); NR (not reported).

## 6. Commercially Available Microfluidic Tests for SARS-CoV-2

The ongoing COVID-19 outbreak has led to high demand for diagnostic tests and their development; thus, some companies have launched different microfluidic products on the market. These devices can provide a qualitative diagnosis of the presence of nucleic acid [91,92,93,94,95,96,97] or viral antigens [98,99,100] of SARS-CoV-2; however, most of them are highly specialized and exclusive to laboratory use [91,92,93,95,96,97,98,99,100]. In addition, these platforms can be fully automated without requiring sample processing, minimizing contamination or human error due to manual handling [91,92,93,94,96]. Semi-automated modalities can also be found. However, the processing of the sample plays a fundamental role since purification and extraction are required to obtain nucleic acid [95,97] or proteins [98,99] before they are loaded into the equipment.

The microfluidic equipment used to identify SARS-CoV-2 nucleic acid is based on assays using RT-qPCR, multiplex RT-qPCR (identifying pathogens and respiratory viruses), RT-LAMP, and isothermal amplification, and in the presence of SARS-CoV-2. These devices release signals that are directly proportional to the number of viral particles present in the sample, detected by fluorescence, voltage, or colorimetric sensors (Table 4). On the other hand, a few commercial microfluidic tests identify SARS-CoV-2 antigens. These are mainly based on sandwich-type immunoassays, which are released in the presence of SARS-CoV-2 fluorescence, electrochemical signals, or resonance frequency changes (Table 4).

Overall, these diagnostic tests provide results within a few minutes, representing a valuable benefit in contrast to technologies that require a long time from sample processing to obtaining the result, allowing the immediate implementation of preventive measures.

## 7. Limitations and Perspectives

An upward trend related to microfluidics, mainly applied for diagnostics, has recently emerged; unfortunately, most of the microfluidic-based diagnostic devices are in a proof-of-concept or prototype stage, which is why more microchips are reported in the literature than are commercially available. Therefore, there is an evident necessity to mature this technology for its practical application in diagnosing infectious diseases. Such microfluidic devices would involve multidisciplinary expertise, compatibility in biological aspects, and feasible manufacturing at a large scale. For example, the materials used for the fabrication of microfluidic chips must be designed for final applications to achieve the optimal performance; a microfluidic chip that requires functionalizing an antibody to detect a viral antigen will use different materials than those used in a microfluidic device capable of performing qPCR to detect a viral gene.

Commonly used materials for the fabrication of microfluidic chips that allow the detection of nucleic acid, antibodies, and proteins of SARS-CoV-2 are based on glass, silicon, polymers (e.g., PDMS), paper, and metal. With respect to glass, this material is ideal for chemical reactions in extreme conditions. One of the significant advantages of silicon is that it allows the molecular diagnosis of biomolecules from diverse samples. However, its high cost for large-scale production is a disadvantage that hinders its application.

On the other hand, microfluidic assays using polymers, paper, or metal have accessible costs for large-scale production; however, they have other drawbacks. The sample should be selected appropriately in the case of platforms that use polymers since this material can present a high or low capacity to adsorb diverse molecules. In comparison, paper-based microfluidic assays are characterized by low cost and small mechanical resistance. A significant limitation is that if the microfluidic device uses passive pumping, it can be a considerable challenge to design the fluid circuit’s hydrodynamic resistance properly. Microfluidic devices made from metal are incompatible with optical detection [102].

Therefore, for large-scale production, it is important to consider the materials to be used, since the transition from the “proof-of-concept” stage to the “final product” stage, especially in microfluidics, can result in drastic changes to the original microfluidic chip design, such as modifications to microfluidic chip materials, dimensions, and structures. Profitability and large-scale manufacturing capacity are the main factors involved in launching a microfluidic product to the market, mainly when it is intended to be widely distributed in a pandemic situation.

Another important factor is the type of biomolecules to be detected. For example, microfluidic assays that detect SARS-CoV-2 nucleic acid have advantages such as high sensitivity compared to microfluidic platforms for antigens or antibodies, requiring up to two viral copies and volumes up to 10 nL due to the exponential amplification [24,25], indicating the presence of the virus in the early stages of infection [103]. These microfluidic devices are the market leaders; however, their high costs compared to immunoassays and the fast degradation of nucleic acids if the samples are not processed immediately or properly stored represent significant disadvantages.

Microfluidic assays for SARS-CoV-2 antigens are used due to the excellent stability that the proteins present compared with the nucleic acid. This type of microfluidic assays can detect SARS-CoV-2 proteins in concentration ranges from fg/mL to µg/mL [57,64].

To avoid false negatives, the proper detection of SARS-CoV-2 proteins should be performed approximately on the fifth day after the infection, which is typically accompanied by tests of nucleic acid to confirm the results [49,50]. On the other hand, microfluidic platforms based on immunoassays to detect antibodies identify the severity of the disease depending on age and levels of expression of anti-SARS-CoV-2 antibodies. They also allow the identification of asymptomatic patients and the evaluation of or the success of vaccination against SARS-CoV-2 with LODs of ng/mL. However, they cannot provide early diagnosis since the production of antibodies takes approximately two weeks post-inoculation [70,71].

The commercially available microfluidic equipment is based on the analysis of nucleic acid or viral antigens of SARS-CoV-2. Currently, there are no commercial microfluidic platforms for identifying anti-SARS-CoV-2 antibodies, for which ELISA is the main method used [90]. In addition, most microfluidic devices are highly specialized and require laboratories to handle them, making them inaccessible in rural areas. In this manner, a significant limitation is the transportation of samples from the collection site to the laboratories, which can cause degradation of the nucleic acid or proteins.

Therefore, the different microfluidic platforms used for diagnosing SARS-CoV-2 and seroprevalence are equally important and complementary to each other, since each one offers valuable results that will indicate the presence or absence of the virus or the state of health of the patient. Furthermore, these devices require similar test times for sample processing and obtaining results and their costs will depend on the material used for their design.

To achieve progress in microfluidic diagnosis, it is also fundamental to consider that some microfluidic chips require pumps or fluid controllers to supply the necessary reagents, which are specialized instruments that many laboratories do not possess. Therefore, one of the main approaches regarding microfluidics in diagnosis is to simplify or avoid using specialized or expensive equipment. This will lead to a greater adaptation of microfluidic systems in different research groups worldwide, improving accessibility to this technology. Consequently, a greater understanding of microfluidics would broaden its applications. The microfluidic proposals applied to COVID-19 reviewed in this work show that a search is evidently occurring in the scientific community for means of reducing test times and costs, improving sensitivity, and simplifying or automating the processing of a test through the employment of microfluidics.

## 8. Conclusions

This review article describes the diagnostic tools that have emerged to detect nucleic acid, viral antigens, and anti-SARS-CoV-2 antibodies, using several techniques such as PCR, isothermal amplification, CRISPR, and immunoassays on microfluidic platforms. Most of these devices are innovative and portable, with the capacity to be stored for months, allowing them to reach communities or places without access to laboratories or hospitals. Furthermore, the microfluidic tests reviewed in this work use minimal amounts of reagents, reducing the total cost and the need for highly qualified personnel. Compared to conventional techniques, microfluidic tests can take minutes to a few hours to obtain accurate results, enabling healthcare professionals to implement efficient treatments and more appropriate infection control decisions. Microfluidic technology can gradually change the course of diagnosis in infectious diseases, especially in successive contagious disease outbreaks. COVID-19 has highlighted the importance of having sufficient supplies and technology that can to be applied directly to the clinical area. The pandemic has left us with several interesting proposals for microfluidic chips that may provide solutions in the near future to enable massive diagnosis, replacing qPCR as the gold standard.

## Figures and Tables

**Figure 1 biosensors-12-00179-f001:**
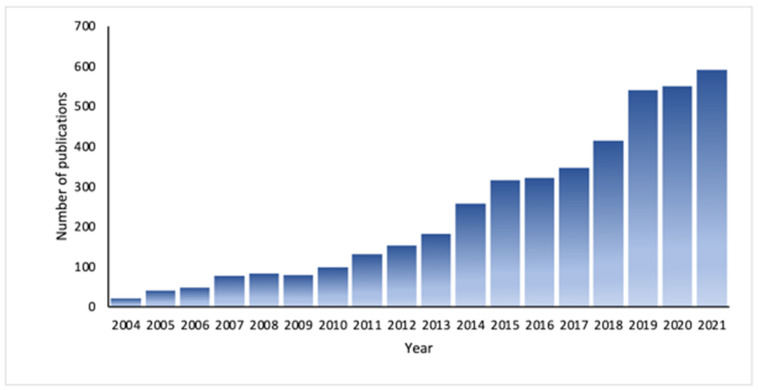
The number of publications per year on microfluidics applied in diagnosis. Note: Scopus database using a combination of the keywords “microfluidics” and “diagnosis”.

**Figure 2 biosensors-12-00179-f002:**
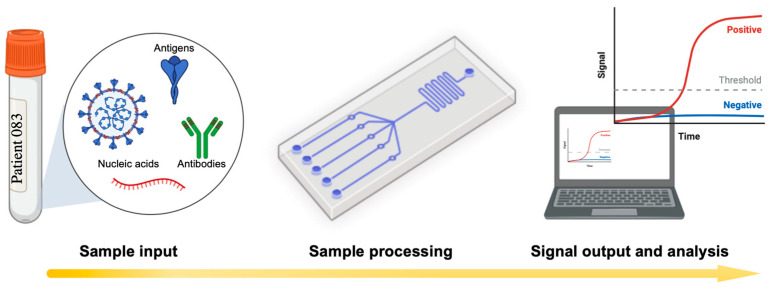
Schematic representation of a basic workflow using a microfluidic platform capable of sensing molecules such as nucleic acid, antigens, and anti-SARS-CoV-2 antibodies, producing a signal output. Created with Biorender.com (accessed on 11 February 2022) and extracted under premium membership.

**Figure 3 biosensors-12-00179-f003:**
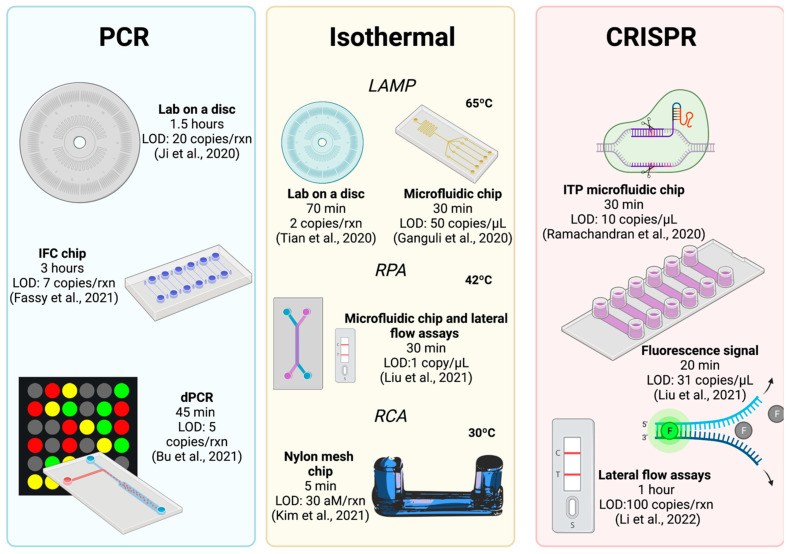
Graphical summary of microfluidic techniques used to detect nucleic acid from SARS-CoV-2 and its main characteristics. Created with Biorender.com (accessed on 11 February 2022) and extracted under premium membership.

**Table 1 biosensors-12-00179-t001:** Microfluidic assays for the detection of SARS-CoV-2 nucleic acid.

Type of Technique	LOD	Target Gene	Detection Method	Processing Time (Minutes)	Reference
qPCR	9 copies/rxn	N	Fluorescence	NR	[23]
qPCR	7 copies/rxn	N, E, ORF1ab, S and NSP6	Fluorescence	<120	[24]
qPCR	7 copies/μL	N	Fluorescence	<120	[25]
qPCR	1 copy/rxn	N	Fluorescence	30	[26]
qPCR	20 copies/rxn	N	Fluorescence	90	[27]
qPCR	259 copies/μL	E	Nanoplasmonic	5	[28]
dPCR	4.68 copies/μL	N and ORF1ab	Fluorescence	45	[31]
dPCR	5 copies/rxn	N and ORF1ab	Fluorescence	5	[32]
dPCR	10 copies/μL	ORF1ab	Fluorescence	<60	[33]
LAMP	2 copies/rxn	N, E and ORF1ab	Fluorescence	70	[35]
LAMP	100 copies/rxn	ORF1ab	Fluorescence	20	[36]
LAMP	50 copies/μL	N, ORF1ab and ORF8	Fluorescence	30	[37]
LAMP	<1 copy/μL	N	Fluorescence	10	[38]
LAMP	200 copies/μL	N and ORF1ab	Colorimetric	60	[39]
LAMP	300 copies/rxn	N and E	Colorimetric	35	[40]
RCA	30 aM/rxn	ORF1ab	Gelation	5	[41]
RPA-LFA	1 copy/μL	N	Colorimetric	30	[42]
RPA-LAMP	10 copies/rxn	S	Fluorescence	60	[43]
CRISPR	10 copies/μL	N and E	Fluorescence	40	[44]
CRISPR-LFA	100 copies/rxn	N	Colorimetric	NR	[45]
CRISPR	31 copies/μL	N, S and ORF1ab	Fluorescence	20	[46]

Abbreviations: LOD (limit of detection); NR (not reported); rxn (reaction); LFA (lateral flow assay).

**Table 4 biosensors-12-00179-t004:** Commercially available microfluidic tests for SARS-CoV-2.

Product	Manufacturer Name	Type of Platform	Target	Detection Method	Processing Time (Minutes)	Reference
ePlex SARS-CoV-2 Test	GenMark Diagnostics, Inc.	RT-qPCR	Nucleic Acid	Voltage	~120	[92]
BioFire COVID-19 test	BioFire Defense, LLC	Multiplex RT-qPCR	Nucleic Acid	Fluorescence	50	[93]
QIAstat-Dx Respiratory SARS-CoV-2 panel	QIAGEN GmbH	Multiplex RT-qPCR	Nucleic Acid	Fluorescence	~60	[94]
Lucira COVID-19 All-In-One Test Kit	Lucira Health, Inc.	RT-LAMP	Nucleic Acid	Colorimetric	30	[95]
Respiratory Virus Nucleic Acid Detection kit	CapitalBio Technology	Isothermal amplification	Nucleic Acid	Fluorescence	90	[96]
Xpert Xpress SARS-CoV-2 test	Cepheid	RT-qPCR	Nucleic Acid	Fluorescence	45	[97]
Microchip RT-PCR COVID-19 detection system	Lumex Instruments Canada	RT-qPCR	Nucleic Acid	Fluorescence	50	[98]
Omnia SARS-CoV-2	Qorvo Biotechnologies	Antigen immunoassay	Proteins	Resonance frequency	~20	[99]
LumiraDx SARS-CoV-2 Ag test	LumiraDx	Antigen immunoassay	Proteins	Fluorescence	12	[100]
Sampinute COVID-19	Celltrion	Antigen immunoassay	Proteins	Electrochemical	30–45	[101]

## Data Availability

Not applicable.

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
