# Peer review of "Microfluidics-Based Biosensing Platforms: Emerging Frontiers in Point-of-Care Testing SARS-CoV-2 and Seroprevalence"

_biosensors, 2022, doi:10.3390/bios12030179_

Round 1

Reviewer 1 Report

The author presented a review of different microfluidic platforms for the diagnosis of COVID-19. The topic is interesting and they have referred to a good list of recently published works. 

Author Response

Dear Reviewer, We highly appreciate your comments and your time and effort in improving this manuscript.

Reviewer: The author presented a review of different microfluidic platforms for the diagnosis of COVID-19. The topic is interesting and they have referred to a good list of recently published works.

Author’s Response: Thanks for your nice overview and useful comments.

Reviewer 2 Report

See Word file

Author Response

Dear Reviewer,

We really appreciate all your effort, time, comments and suggestions!!!

Reviewer: The authors reviewed microfluidics-based biosensing platforms for the detection of COVID-19. The authors give a nice overview and the article has scientific value. Therefore, I would advise to publish this paper, after some minor revisions. In particular:

Author’s Response: Thanks for your nice overview and useful comments. The whole manuscript has been revised and updated following all given comments and suggestion. A point-by-point response to each comment is also given below for Reviewer’s and Editor’s further consideration:

Comments:

  1. I would change the title of the article. With the mentions microfluidic devices, the analyte SARS-CoV-2 virus is detected and not the illness COVID-19.

Author’s Response: We appreciate this suggestion, and we are totally agreed, we have changed the title accordingly.

  1. Why is for the antibody detection anti-SARS-CoV-2 specifically mentioned every time and not for the nucleic acids and antigens? I would remove anti-SARS-CoV-2, since it is clear from the context that is about SARS-CoV-2 detection.

Author’s Response: Thanks for your comment. In the case of the antibodies, this is the proper format to refer to virus-specific antibodies. Please find below scientific articles using the same notation.

https://bmcinfectdis.biomedcentral.com/articles/10.1186/s12879-021-06550-5

https://www.eurosurveillance.org/content/10.2807/1560-7917.ES.2021.26.43.2100830

https://pubmed.ncbi.nlm.nih.gov/34611927/

https://jasn.asnjournals.org/content/32/5/1033

Check the lay-out and grammar of the document, some examples:

  1. R 21: remove the word ‘also’ (there is already ‘moreover’ in the sentence)

Author’s Response: We highly appreciate your comments and your time and effort in improving this manuscript. All your suggestions have been considered and the corresponding modifications have been made.

  1. Missing spaces between sentences (e.g. r 44, r 84)

Author’s Response: Thanks for noticing this, it has been corrected.

  1. Two spaces between sentences (e.g. r 423)

Author’s Response: Thanks for noticing this, it has been corrected.

  1. R 106 is wrongfully in the past tense

Author’s Response: Thanks for noticing this, it has been corrected.

  1. R 116-117: helped to diagnose

Author’s Response: Thanks for noticing this, it has been corrected.

  1. R 118: reagent(s) (instead of reactive)

Author’s Response: Thanks for noticing this, it has been corrected.

  1. R 134: price of a test or test price

Author’s Response: Thanks for noticing this, it has been corrected.

  1. R 155: mention the name who reported this

Author’s Response: Thanks for noticing this, it has been corrected.

  1. R165: rephrase the sentence after the ‘;’

Author’s Response: Thanks for noticing this, it has been corrected.

  1. R175: something is missing in this sentence (names?)

Author’s Response: Thanks for noticing this, it has been corrected.

  1. R206: minutes instead of min

Author’s Response: Thanks for noticing this, it has been corrected.

  1. R240: detection limit or limit of detection

Author’s Response: Thanks for noticing this, it has been corrected.

  1. R274: SARS-CoV-2

Author’s Response: Thanks for noticing this, it has been corrected.

  1. R291: there are a few reports

Author’s Response: Thanks for noticing this, it has been corrected.

  1. N instead of NP is the abbreviation of nucleocapsid (NP is also the common

abbreviation for nasopharyngeal)

Author’s Response: Thanks for noticing this, it has been corrected.

  1. R436: fluorescecent signal

Author’s Response: Thanks for noticing this, it has been corrected.

  1. Table 2: Cepheid

Author’s Response: Thanks for noticing this, it has been corrected.

  1. Make a reference in the text to the device if it is depicted in Figure 3.

Author’s Response: Thanks for noticing this, it has been corrected.

  1. The lay-out of table 1 needs to be checked.

Author’s Response: Thanks for noticing this. As suggested, it has been modified.

  1. In general, the sensitivity of the devices is mentioned. But the selectivity is as important as the sensitivity. Mention this as well (or indicate that it is not determined, if that is the case).

Author’s Response: Thanks for noticing this. As suggested, it has been modified.

  1. The discussion (section 7) is too brief. Elborate more on the discussed devices and give a critical reflection (see also the feedback for this section)

Author’s Response: We appreciate this recommendation. As suggested, we have extended the section 7 with a critical reflection of the different types of microfluidics devices for the detection of SARS-CoV-2.

  1. Microfluidics applied on COVID-19. I would rather say SARS-CoV-2 (detection) instead of COVID-19.

Author’s Response: Thanks for this suggestion, it has been corrected.

  1. In the caption of Figure 1 it is stated that keywords ‘such as’ are used. I would like to know which exact keywords are used in order to make Figure 1.

Author’s Response: Thanks for noticing this. The exact keywords are “microfluidics” and “diagnosis”. 

  1. Nucleic acid detection. Nucleic acid detection instead of nucleic acids detection

Author’s Response: Thanks for noticing this, it has been corrected.

  1. What are the concentrations (number of copies per volume) for the mentioned devices in Figure 3?

Author’s Response: Thanks for noticing this, it has been added.

  1. Add a table (like table 1) with the characteristics of nucleic acid detection devices

Author’s Response: Thanks for noticing this, it has been added.

  1. R162: what is meant with low-density samples?

Author’s Response: Thanks for noticing this, it has been corrected.

  1. LAMP makes use of 4 to 6 primers (not 8)

Author’s Response: Thanks for noticing this, it has been corrected.

  1. R221: mention the name of the research group (like everywhere else)

Author’s Response: As suggested, the name of the research group has been included.

  1. If possible, use everywhere the same units for the LOD: R228: how much, in copies/μL, is 30 aM? R289: again the question, how much is 250 amol/mL in copies/μL?

Author’s Response: We appreciate this observation. However, we decided to report the units as presented in the revised articles because the difficulties on the conversion can lead to errors. These difficulties are associated to the fact that researchers use different templates and frequently they do not report relevant data for a proper conversion. Thus, we consider that units should be maintained as presented in their articles, to avoid erroneous data.

  1. Anti-SARS-CoV-2 antibodies detection. This is again an extensive summary of articles, whereby you easily get lost in all the numbers and names. Add a table (like table 1) with the characteristics of antibody detection devices.

Author’s Response: Thanks for noticing this, it has been added

  1. Commercially available microfluidic tests for COVID-19. I would rather say SARS-CoV-2 (detection) instead of COVID-19.

Author’s Response: It has been modified as suggested.

  1. Elaborate somewhat more on the commercially available devices.

Author’s Response: We appreciate this recommendation. Section 6 (commercially available microfluidic tests for SARS-CoV-2 was extended).

  1. Limitations and perspectives. The materials are mentioned as parameter for optimal performance. Elaborate on what chip materials are possible and the pros and cons of each material.

Author’s Response: Thanks for this recommendation. We have added a detailed discussion regarding the advantages and disadvantages presented depending on the selection of the materials.

  1. Eleborate on which type of detection is the most sensitive, most fast, uses the smallest volume, is the cheapest, etc.

Author’s Response: As suggested, we have added a comparative analysis of the different types of detection 

  1. When do you want to use a nucleic acid test and when an antigen test? Elaborate on the different types of tests.

Author’s Response: Thanks for this recommendation. As suggested we have added this discussion 

  1. Why are there many developments within academics/published in papers, but so few commercially available devices (at least for nucleic acid detection (of SARS-CoV-2))?

Author’s Response: Thanks for this recommendation. As suggested we have added this discussion

Reviewer 3 Report

To deal with the current healthcare crisis, many microfluidic platforms have been developed for diagnosing the SARA-COV-2 virus which causes the coronavirus disease (COVID-19). In this manuscript, González-González et al., summarized the state-of-art of the development of microfluidic biosensing platforms for point-of-care COVID-19 detection at the level of nucleic acids, antigens, and anti-SARS-CoV-2 antibodies. The manuscript is comprehensive and well-written. With the following issues being addressed, the reviewer would suggest its publication in Biosensors.

  1. The tense should be consistent. For example, Line 17, “In this review, we discuss the different microfluidic platforms applied in the diagnosis of COVID-19, which were classified into three sections according to the molecules to be detected: nucleic acids, antigens, and anti-SARS-CoV-2 antibodies. Moreover, commercially available alternatives based on microfluidic platforms were also described.”
  2. The Figures are informative and clearly summarize the research progresses and methodologies.
  3. The link for reference 34 is not correct. It should be this one. https://pubs.rsc.org/en/content/articlehtml/2021/an/d0an02066d
  4. The reviewer would suggest to discuss the advantage and limitation of these commercially available microfluidic tests for COVID-19 to give an in-depth comparison.
  5. There are some typos, such as Line 68, “Microfluidic is the basis of technologies such as labs-on-a-chip or micro total analysis”. Line 79, “Microfluidics diagnostic systems are integrated by detection components and fluid regulatory elements in microenvironments [15] able to detect different molecules such as”. Line 257, “samples and simultaneously mixed, preconcentrate, and accelerate enzymatic reactions”.

Author Response

Dear Reviewer,

Thanks for your useful comments. The whole manuscript has been revised and updated following all given comments and suggestion.  

Comments:

  1. The tense should be consistent. For example, Line 17, “In this review, we discuss the different microfluidic platforms applied in the diagnosis of COVID-19, which were classified into three sections according to the molecules to be detected: nucleic acids, antigens, and anti-SARS-CoV-2 antibodies. Moreover, commercially available alternatives based on microfluidic platforms were also described.”

Author’s Response: We appreciate this suggestion. It has been modified as recommended.

  1. The Figures are informative and clearly summarize the research progresses and methodologies.

Author’s Response: We appreciate your nice comments.

  1. The link for reference 34 is not correct. It should be this one. https://pubs.rsc.org/en/content/articlehtml/2021/an/d0an02066d

Author’s Response: Thanks for noticing this. It has been corrected.

  1. The reviewer would suggest to discuss the advantage and limitation of these commercially available microfluidic tests for COVID-19 to give an in-depth comparison.

Author’s Response: We appreciate this recommendation. Section 6 (commercially available microfluidic tests for SARS-CoV-2 was extended in L788-L828.

  1. There are some typos, such as Line 68, “Microfluidic is the basis of technologies such as labs-on-a-chip or micro total analysis”. Line 79, “Microfluidics diagnostic systems are integrated by detection components and fluid regulatory elements in microenvironments [15] able to detect different molecules such as”. Line 257, “samples and simultaneously mixed, preconcentrate, and accelerate enzymatic reactions”.

Author’s Response: Thanks for noticing this. The whole manuscript was carefully revised and corrected.

Reviewer 4 Report

The current review article, ' Microfluidics-based biosensing platforms: Emerging frontiers in point-of-care testing COVID-19' by Flores-Contreras et al. is an informative and updated review article that includes recent developments in microfluidics research in deteting covid-19 viruses. I recomend it to be published after including the following minor revisions to the article.

  1. The quality of English language in the article is poor. I strongly recomend a throrough corrections of English and recheck by a professional English scientific article writer. Start corrections in English from the title itself to the end of conclusion section! 
  2. Maintain the same pattern of references, e.g., ref. 42 does not include the page numbers! Check all the references and correct them, if needed.
  3. The authors highlighted some of the major spectroscopic techniques including fluorescence, FT-IR. In recent decades, Raman and SERS found major applications in detecting biological target antigens and nucleic acid sequences. I recomend the authors to include atleast one report of Raman or SERS based detection of covid-19 antigens and nucleic acid.

Author Response

Dear Reviewer,

Thanks for your useful comments  

Reviewer #4: The current review article, ' Microfluidics-based biosensing platforms: Emerging frontiers in point-of-care testing COVID-19' by Flores-Contreras et al. is an informative and updated review article that includes recent developments in microfluidics research in deteting covid-19 viruses. I recomend it to be published after including the following minor revisions to the article.

Author’s Response: Thanks for your nice overview and useful comments. The whole manuscript has been revised and updated following all given comments and suggestion. A point-by-point response to each comment is also given below for Reviewer’s and Editor’s further consideration:

Comments:

  1. The quality of English language in the article is poor. I strongly recomend a throrough corrections of English and recheck by a professional English scientific article writer. Start corrections in English from the title itself to the end of conclusion section!

Author’s Response: We appreciate this recommendation. The whole manuscript was carefully revised and corrected.

  1. Maintain the same pattern of references, e.g., ref. 42 does not include the page numbers! Check all the references and correct them, if needed.

Author’s Response: Thanks for noticing this. We have revised the references and corrected them as suggested.

  1. The authors highlighted some of the major spectroscopic techniques including fluorescence, FT-IR. In recent decades, Raman and SERS found major applications in detecting biological target antigens and nucleic acid sequences. I recomend the authors to include atleast one report of Raman or SERS based detection of covid-19 antigens and nucleic acid.

Author’s Response: Thanks for noticing this, it has been added.